Volumetric atlas of the rat inner ear from microCT and iDISCO+ cleared temporal bones

Cossellu Daniele 1
Vivado Elisa 1
Batti Laura 2
http://orcid.org/0000-0001-6351-8665 Gantar Ivana 2
http://orcid.org/0000-0003-4884-8265 Pizzala Roberto 1 3 rpizzala@unipv.it
http://orcid.org/0000-0002-5897-4444 Perin Paola 3 4
1 Department of Molecular Medicine, University of Pavia , Pavia , Italy
2 Wyss Center for Bio and Neuro Engineering , Geneva , Switzerland
3 Fondazione IRCCS Policlinico San Matteo , Pavia , Italy
4 Department of Brain and Behavioral Sciences, University of Pavia , Pavia , Italy
Stefen Clara
Electronic publication date: 2025 May 26
Publication date: 2025
Volume: 13
Electronic Location ID: e19512
Received 2024 Feb 2; Accepted 2025 May 2
Copyright: © 2025 Cossellu et al.
Copyright year: 2025
Copyright holder: Cossellu et al.
License: This is an open access article distributed under the terms of the Creative Commons Attribution License, which permits unrestricted use, distribution, reproduction and adaptation in any medium and for any purpose provided that it is properly attributed. For attribution, the original author(s), title, publication source (PeerJ) and either DOI or URL of the article must be cited.
License URL: https://creativecommons.org/licenses/by/4.0/

Keywords: Inner ear anatomy, Rat animal model, iDisco+, Immunofluorescence, Micro CT, Tissue clearing, Inner ear, 3D atlas, Lightsheet microscopy

Funding: Banca del Monte di Lombardia Foundation, contribution to Prof. Roberto Pizzala, Dipartimento di Medicina Molecolare, University of Pavia 2021 This work was supported by the Banca del Monte di Lombardia Foundation, contribution to Prof. Roberto Pizzala, Dipartimento di Medicina Molecolare, University of Pavia 2021. The funders had no role in study design, data collection and analysis, decision to publish, or preparation of the manuscript.

==============================
Background

Volumetric atlases are an invaluable tool in neuroscience and otolaryngology, greatly aiding experiment planning and surgical interventions, as well as the interpretation of experimental and clinical data. The rat is a major animal model for hearing and balance studies, and a detailed volumetric atlas for the rat central auditory system (Waxholm) is available. However, the Waxholm rat atlas only contains a low-resolution inner ear featuring five structures. In the present work, we segmented and annotated 34 structures in the rat inner ear, yielding a detailed volumetric inner ear atlas which can be integrated with the Waxholm rat brain atlas.

Methods

We performed iodine-enhanced microCT and iDISCO+-based clearing and fluorescence lightsheet microscopy imaging on a sample of rat temporal bones. Image stacks were segmented in a semiautomated way, and 34 inner ear volumes were reconstructed from five samples. Using geometrical morphometry, high-resolution segmentations obtained from lightsheet and microCT stacks were registered into the coordinate system of the Waxholm rat atlas.

Results

Cleared sample autofluorescence was used for the reconstruction of most inner ear structures, including fluid-filled compartments, nerves and sensory epithelia, blood vessels, and connective tissue structures. Image resolution allowed reconstruction of thin ducts (reuniting, saccular and endolymphatic), and the utriculoendolymphatic valve. The vestibulocochlear artery coursing through bone was found to be associated to the reuniting duct, and to be visible both in cleared and microCT samples, thus allowing to infer duct location from microCT scans. Cleared labyrinths showed minimal shape distortions, as shown by alignment with microCT and Waxholm labyrinths. However, membranous labyrinths could display variable collapse of the superior division, especially the roof of canal ampullae, whereas the inferior division (saccule and cochlea) was well preserved, with the exception of Reissner’s membrane that could display ruptures in the second cochlear turn. As an example of atlas use, the volumes reconstructed from segmentations were used to separate macrophage populations from the spiral ganglion, auditory neuron dendrites, and Organ of Corti.

Conclusion

We have reconstructed 34 structures from the rat temporal bone, which are available as both image stacks and printable 3D objects in a shared repository for download. These can be used for teaching, localizing cells or other features within the ear, modeling auditory and vestibular sensory physiology and training of automated segmentation machine learning tools.

Introduction

The inner ear is appropriately named “labyrinth” given its complex geometrical structure. For this reason, one of the main endeavors of inner ear research has been the description of its parts, which has grown progressively more precise with the availability of novel tools. The 3D reconstruction of inner ear structures in situ is now possible thanks to nondestructive imaging methods based on X-rays and fluorescence (see Vogl, Neef & Wichmann, 2022 for an extensive review) and software tools for morphometrical analysis (Rolfe et al., 2020; Porto, Rolfe & Maga, 2021). Precise 3D reconstructions provide a geometrical framework for finite element modeling (Castle et al., 2023), cochlear implant positioning (Keppeler et al., 2021), performing in situ cochleograms (Santi et al., 2004; Hutson et al., 2021) or testing clinical hypotheses, e.g., on fluid pathways (Brown et al., 2016) and otolith metabolism (Barozzi et al., 2013). In addition, morphometrical tools allow a quantification of shapes (Landi & O’Higgins, 2019) which is essential for comparison between species (Ekdale, 2013), developmental phases (Bryant et al., 2021) and pathological conditions (Santi & Johnson, 2022). For most uses, both X-rays and fluorescence-based imaging methods produce similar results; the latter however also allows selective labeling of soft tissue components (Buytaert et al., 2013), which is essential for understanding physiological and pathological processes such as, for example, fibrosis after cochlear implantation (Brody et al., 2020) or immune cell distribution and interactions (Perin et al., 2019; Urata & Okabe, 2023).

Temporal bone clearing has a long history, having been pioneered by Spalteholz, which made tissues transparent by immersion in a mixture of methylsalicylate and benzyl benzoate (Spalteholz, 1911). This technique was classically employed for detecting inner ear vascular connections (Rask-Andersen, 1979; Mazzoni, 1990) and was improved by the group of Santi (2011), who developed a special microscope for the imaging of the intact cleared cochlea (OPFOS). More recently, tissue clearing has become available to many groups, thanks to the availability of general-purpose clearing methods (Ueda, Ertürk & Chung, 2020) lightsheet microscopes (Santi & Johnson, 2022; Ueda et al., 2020) and both commercial and open-source image analysis software (see an overview of available software tools for image analysis in Haase et al., 2022). Soft tissue reconstructions of the cleared cochlea from mouse (MacDonald & Rubel, 2008; Buytaert et al., 2013) and gerbil (Risoud et al., 2017; Hutson et al., 2021) have been published, and for the guinea pig a reconstruction of the whole labyrinth from lightsheet microscopy imaging (Hofman, Segenhout & Wit, 2009) and iodine-contrasted microCT (Lee et al., 2010) is available. On the other hand, no soft tissue volumetric data are available for the inner ear of the rat.

In hearing and balance research, as in other neuroscience fields, animal models are essential for the understanding of physiology and pathology; mice and rats are the most used species (Bryda, 2013), although other models may offer unique advantages for selected uses (reviewed in Straka, Zwergal & Cullen (2016), Castaño-González, Köppl & Pyott (2024)). Auditory and vestibular physiology is as well characterized in rat as in mouse (reviewed in Albuquerque et al. (2009), Alvarado et al. (2014), Escabi et al. (2019), Holt et al. (2019)), and the rat genome has been completely sequenced (Rat Genome Sequencing Project Consortium et al., 2004). Moreover, a large number of antibodies are available for immunolabeling in rat.

In the last decades, the use of rats in inner ear research has lagged behind mice, mainly because a very large database of mouse mutants is available for genetic studies (Blake et al., 2021), including hearing-related ones (Bowl et al., 2017; Lewis et al., 2022), whereas genetically modified rats have been obtained only much later (Chenouard et al., 2021) and very few transgenic rat models are available for hearing or balance phenotypes (Auerbach et al., 2021; Gansemer et al., 2024). However, rats have some intrinsic advantages when compared to mice, which include greater robustness and larger size, which facilitates surgery and tissue harvesting, and greater sociability and behavioral flexibility, which makes them easier to train in behavioral protocols (Bryda, 2013). Moreover, the human immune system is more similar to rats than mice (Wildner, 2019) and the rat may therefore better help understanding immune-mediated inner ear diseases (Gansemer et al., 2024). In hearing research, rats are still an animal model of choice for physiology, toxicology, and behavioral neuroscience (Jacob, 2010; Jaramillo & Zador, 2014), and represent a major model in central auditory neuroscience (Malmierca, 2003).

A complete volumetric atlas of the rat central auditory system has been generated (Osen et al., 2019) and integrated in the Waxholm Space atlas of the Sprague Dawley rat brain (Papp et al., 2014; Kjonigsen et al., 2015; Osen et al., 2019; Kleven et al., 2023). A low-resolution scan of the cochlea, vestibular labyrinth, and relative nerves is part of this atlas, but is too coarse to resolve most inner ear structures. A higher resolution volumetric atlas of the inner ear could help integrating the observations of inner ear properties with those of the brain: e.g., it could be possible to obtain a virtual cochleogram by segmenting the Organ of Corti or quantify the spiral ganglion neuron changes after an ototoxic insult, while also observing central auditory structures. The use of iDISCO+ (immunolabeling-enableD Imaging of Solvent-Cleared Organs; Renier et al. (2014), a clearing technique commonly used for the brain, combined with bone decalcification, allows the parallel observation of both central and peripheral structures (Perin et al., 2019), therefore no additional treatments except decalcification are needed for observing the inner ear together with the brain. Moreover, since head scans with nondestructive imaging techniques very often contain the cranium and may display interesting features in the inner ear (e.g., see Extended Data Figure 1i in Ahn et al. (2019)) adding an annotated inner ear section to the 3D rat brain atlas seems timely.

This volumetric atlas is intended for two main audiences with different levels of inner ear expertise. The first group consists of comparative anatomy educators and students (Kapoor, 2024), who may not have a clear picture of the organization of the inner ear in rodents. In fact, although several very good quality volumetric reconstructions of the inner ear have been produced in the literature, unfortunately in most cases only the article is accessible, due to infrastructure lability leading to loss of data repositories (Strecker et al., 2023). For educational purposes, having an open source, high-resolution, printable volumetric model of the whole inner ear and its parts represents a useful tool to understand the organization of the sensory organs. The second group includes neuroscientists which model auditory and vestibular physiology. For research purposes, this atlas yields quantitative measurements of the rat inner ear, which are useful for setting parameters e.g., in computational models or for microsurgical procedures, plus an annotated dataset useful for localizing labeling within the inner ear regions.

Materials and Methods

Experiments were performed on adult inbred Wistar rats of both sexes (Table 1). Five temporal bones from four rats were used for whole inner ear reconstruction, and two additional samples were employed to help resolving segmentation details.

Table 1 List of samples used for segmentation.

Sample	Age (days)	Sex	Signal	Voxel size	
A	92	M	Auto	4.08	
B	494	M	Auto, SMA	3.26	
C1	79	F	uCT-Lugol	6.23	
C2	79	F	uCT-Lugol	6.23	
D	111	F	Iba-1	2	
E	79	M	ToPro	3	
F	92	F	Auto	2	
Note:

Animals, ages, sex, voxel size and labeling are shown.

This study was carried out in accordance with the recommendations of Act 26/2014, Italian Ministry of Health. The protocol (number 155/2017-PR) was approved by the Italian Ministry of Health and University of Pavia Animal Welfare Office (OPBA). All efforts were made to minimize the number of animals used and animal suffering.

Image stacks used for this work come from the same samples studied for the reconstruction of temporal bone marrow (Perin et al., 2024): therefore, all sample treatments were as in Perin et al. (2024). Tissue clearing was performed with a variant of iDISCO+ (Renier et al., 2014) which allowed to image both brain and temporal bone (Perin et al., 2019). Decalcification was passive (30d in EDTA 5% with daily changes) except for rat B where it was microwave-aided (8 h for 3 days at 40 °C, with a change every hour, using Milestone MLS 1200 Mega high-performance microwave). Fluorescent labeling of cleared samples was performed as in Perin et al. (2024); the antibodies used in the present article were: mouse anti-smooth muscle actin (SMA; Abcam, amab7817, 1:200) and rabbit anti-Iba1 (WAKO 019–19,741, 1:200); secondary antibodies were donkey anti-rabbit and anti-mouse conjugated with Alexa 488, 555 or 647 (Invitrogen, 1:200). One sample was also counterstained with the nuclear dye TO-PRO (Sigma T3605, 1: 500 or 1:1000). Cleared samples were imaged with a mesoSPIM lightsheet microscope (Voigt et al., 2019) at the Wyss Center for Bio and Neuroengineering in Geneva, Switzerland, as in Perin et al. (2024). To reduce blur, lightsheet image stacks were deconvolved using the MATLAB script as in Becker et al. (2019), with a NA of 0.14 (Voigt et al., 2019) and the refractive index of DBE (1.562). This deconvolution algorithm applies a correction to lightsheet images using a theoretical lightsheet PSF (point spread function) model, thus increasing image sharpness. After deconvolution, image stacks were rescaled to isotropic with FIJI (voxel sizes are given in Table 1).

MicroCT samples (rat C, both sides) were counterstained with iodine by immersion in Lugol’s iodine (KI3) solution (Metscher, 2009) until solution clearing, for soft tissue staining. MicroCT observations were performed with a SkyScan 1276 CMOS (Bruker, Kontich, Belgium) at the University of Pavia core facility, using step and shoot acquisition with a 0.2-degree rotation step. Images were reconstructed using the software Bruker Nrecon 1.7.4 with ring artefact correction and beam hardening, and Gaussian smoothing to limit rogue voxel artefacts (Nichols & Holmes, 2007). The sample was observed with source settings of 85 kV/47 μA and 1 mm aluminum filter. Images were 3,272 × 3,092 × 2,189 pixels, with an isotropic 6.4 μm3 voxel size.

Image analysis

Segmentation

Segmentation of the inner ear labyrinth from image stacks was performed in a semiautomated way using FIJI (Schindelin et al., 2012) and ITK-SNAP (Yushkevich, Gao & Gerig, 2016) with the following pipeline: (1) CLAHE (FIJI macro)

(2) raw threshold segmentation with Otsu (FIJI macro)

(3) generation of initial segmentation mask (FIJI macro)

(4) gross manual correction of mask (FIJI)

(5) snake segmentation optimization (ITK-SNAP) using mask as seed

(6) manual correction of segmentation (ITK-SNAP)

(7) iterate 5-and 6 until optimal segmentation

Snake segmentation, also called active contour segmentation (Yushkevich et al., 2006), extracts image boundaries in the presence of heterogeneities of brightness and contrast. This approach does not employ thresholding but, starting from an initial selected region (or “seed”), minimizes an “energy” function which affects the speed at which region boundary points move within the image, expanding or shrinking the segmented region. For each boundary point, the energy function is affected by parameters related to local image brightness gradient and boundary curvature, which can be manually adjusted.

Three expert segmentators performed independent segmentations and volumes were considered acceptable when volume overlap was >90%

Segmentations of iDISCO+ cleared samples were based on the autofluorescence signal at 488 and 647 nm, as described in Perin et al. (2024). In addition, arteries were segmented using SMA immunofluorescence. Boundaries of adjacent structures were added or adjusted by manual segmentation and comparison with other stacks, where signal ambiguities arose. The criteria used for identification and delineation of boundaries are described in detail for each region in the Results section.

Segmentation yielded volumes for 34 structures (Table 2), which were saved as NIfTI image stacks with text labels, and converted into .vtk files for registration, and to .ply files for 3D printing or display.

Table 2 List of segmented structures with colors and abbreviation legend.

Where available, structures are compared to those found in the Waxholm atlas of the rat auditory system (Osen et al., 2019). Colors given here are used throughout the paper except where specific details were highlighted.

Structure	Abbreviation	Color (HEX)	Comparison with Osen et al. (2019)	
Inner ear*1	IE	#00ff00	NA	
Membranous labyrinth*1	Mel	#ff0000	NA	
Temporal bone	Tbo	#cccccc	NA	
Stapes	Sp	#999999	NA	
Round window membrane	Co,rwm	#ffe4c4	NA	
Cochlear nerve (central)	8cn, c	#fffaaa	Partly replacing Cochlear nerve (121)	
Cochlear nerve (peripheral)	8cn, p	#fff000	Partly replacing Cochlear nerve (121)	
Cochlear nerve (dendrite)	8cn, d	#d2b62a	Partly replacing Cochlear nerve (121)	
Spiral ganglion	8cn, SpG	#dacd43	Partly replacing Cochlear nerve (121)	
Saccule (endo)	VeA, sac	#b95b51	Partly replacing Vestibular apparatus (119)	
Utricle (endo)	VeA, utr	#8c7064	Partly replacing Vestibular apparatus (119)	
Semicircular canal (endo)	VeA, scc	#d8c5c5	Partly replacing Vestibular apparatus (119)	
Endolymphatic duct	VeA, ed	#7a4040	NA	
Reuniting duct	VeA, rd	#7a4040	NA	
Superior vestibular nerve	8vn, s	#ffff00	partly replacing Vestibular nerve (122)	
Inferior vestibular nerve	8vn, i	#fffbc9	partly replacing Vestibular nerve (122)	
Cochlear Scala tympani	Co, st	#00ff00	Partly replacing Cochlea (120)	
Cochlear Scala vestibuli*2	Co, sv	#3d8c3d	Partly replacing Cochlea (120)	
Cochlear Scala media	Co, sm	#e69595	Partly replacing Cochlea (120)	
Cochlear Stria vascularis	Co, sva	#ff0000	Partly replacing Cochlea (120)	
Cochlear spiral ligament	Co, slg	#00ffff	Partly replacing Cochlea (120)	
Cochlear spiral limbus	Co, slm	#b79cdc	Partly replacing Cochlea (120)	
Cochlear organ of corti	Co, OoC	#cc00ff	Partly replacing Cochlea (120)	
Saccular connective tissue	VeA,csa	#0b8383	NA	
Membrana limitans	VeA, ml	#Ffffff	NA	
Saccular macula	VeA,msa	#Ff00ff	Partly replacing Vestibular apparatus (119)	
Utricular macula	VeA, mut	#0097ce	Partly replacing Vestibular apparatus (119)	
Posterior canal crista	VeA, pcc	#0000ff	Partly replacing Vestibular apparatus (119)	
Anterior canal crista	VeA, pcc	#029eff	Partly replacing Vestibular apparatus (119)	
Lateral canal crista	VeA, lcc	#0000a3	Partly replacing Vestibular apparatus (119)	
Vestibular perilymph	VeA, p	#3d8c3d	Partly replacing Vestibular apparatus (119)	
Cochlear artery	Art, co	#b00000	NA	
Vestibular artery	Art, ve	#ff512e	NA	
Stapedial artery	Art, sp	#ff9090	NA	
Notes:

*1 Derives from the combination of other volumes.

*2 This volume is divided in two parts in rat B files to allow better depiction of the periotic cistern contents.

Registration

We registered our inner ear segmentations to the right labyrinth of the Waxholm rat atlas (version 4.01; Kleven et al., 2023), to allow substituting the current inner ear (which only contains five structures) with our segmentation (which includes 29 inner ear structures, plus 5 correlated structures—arteries and bones). Registration was performed using geometric morphometrics (Chen et al., 2023). In this approach (Fig. S1), landmarks are placed on defined points for each structure, and registering multiple analogous structures (e.g., bony labyrinths from different samples) means minimizing the distance between the landmark of one structure and the centroid of the corresponding landmarks from all structures (Procrustes distance).

Our registration pipeline was as follows: (1) Bony labyrinth volumes from left ears were mirrored, because the Waxholm low-resolution labyrinth used for registration (hereby called volume W) was from the right ear.

(2) Registration of bony labyrinths to volume W was performed using the morphometric analysis SlicerMorph module (Rolfe et al., 2020) of 3DSlicer (Kikinis, Pieper & Vosburgh, 2014). In detail: (a) Pseudolandmarks were automatically generated on volume W;

(b) Pseudolandmarks were transferred to other bony labyrinth volumes using ALPACA (Porto, Rolfe & Maga, 2021);

(c) Bony labyrinths from all samples were registered to volume W using the Fiducial Registration Wizard from the SlicerIGT module (Ungi, Lasso & Fichtinger, 2016), obtaining transform matrices (which define rotation, translation and scaling which minimize Procrustes distances). Segmented structures which had no counterpart in Volume W were registered to it by applying for each sample the same transform matrices obtained for the bony labyrinth, moving them in the position they would occupy in the target volume.

Labels with volume annotations are available as .txt files in Folder S1, coordinates for the landmarks used in registrations generated by SlicerMorph are available as .fcsv files in Folder S2 and transform matrices for affine transformations generated by SlicerIGT are available as .h5 files in Folder S3.

The volumes obtained from rats A-D are given as NIfTI for inner ear structures. Two sets of NIfTI files are given: the originals and the transformed which are correctly scaled and positioned within the WXS coordinates (Kleven et al., 2023). Other samples were used for supplemental anatomical information (see Results) but are not integrated with the atlas.

Measurements

Volumes and lengths

The volumes of all segmented inner ear structures (Table 3) were directly measured as ITK-SNAP voxel counts, scaled to voxel size. Distances (reuniting duct, endolymphatic duct, basilar membrane) were measured manually with 3DSlicer Markups module, by placing markers along the path to be measured.

Table 3 Volumes of segmented structures (mm3) for each sample.

For comparison with literature data obtained in fresh tissue it has to be considered that fixation, Lugol and iDISCO+ clearing all induce shrinking (see main text).

	A	B	C2	C1	D	AVERAGE	
Cochlea, Scala media	0.87	0.67	–	–	0.52	0.69 ± 0.18	
Cochlea, Scala vestibuli	1.54	1.53	–	–	1.69	1.59 ± 0.09	
Cochlea, Scala tympani	1.28	1.02	–	–	0.98	1.09 ± 0.16	
Cochlea, Spiral ligament	–	0.43	–	–	0.45	0.44 ± 0.01	
Cochlea, Stria vascularis	–	0.05	–	–	0.10	0.08 ± 0.04	
Cochlea, Spiral limbus	0.17	0.10	–	–	0.09	0.12 ± 0.04	
Cochlea, Organ of Corti	–	0.05	–	–	0.04	0.05 ± 0.01	
8th nerve, Cochlear		0.32	–	0.31		0.32 ± 0.01	
8th nerve, Cochlear spiral ganglion	–	0.11	–	–	0.09	0.10 ± 0.01	
8th nerve, Cochlear dendrite	–	0.05	–	–	0.05	0.05 ± 0.00	
8th nerve, Cochlear Spiral ganglion canal	–	–	0.13	0.15	–	0.14 ± 0.01	
8th nerve, Vestibular nerve, superior	0.22	0.15	–	–		0.19 ± 0.05	
8th nerve, Vestibular nerve, inferior	0.06	0.07	–	–		0.07 ± 0.01	
Vestibular apparatus, semicircular canals	1.25	–	–	–		1.25	
Vestibular apparatus, utricle	0.20	0.14	–	–	0.12	0.15 ± 0.04	
Vestibular apparatus, saccule	0.12	0.13	–	–	0.06	0.10 ± 0.04	
Vestibular apparatus, saccular connective tissue	0.08	0.06	–	–		0.07 ± 0.01	
Vestibular apparatus, perilymphatic space	1.00	–	–	–		1.00	
Stapes	0.07	0.11	0.06	0.05		0.07 ± 0.03	
Bony labyrinth	6.51		6.90	7.05	6.96	6.86 ± 0.24	

Reuniting duct

Since no unequivocal boundary was observed for reuniting and endolymphatic ducts, their extremities were defined as follows (Fig. S2). The reuniting duct was considered to start where the scala media opened up behind the spiral limbus, changing direction rather abruptly, and to end at the saccular base, where a plane perpendicular to the saccular membrane could separate reuniting and endolymphatic ducts. The latter was defined to start where the sacculus base started to funnel. Duct sizes were measured manually with 3DSlicer Markups module; since both ducts were flattened ovals, largest and smallest diameter values are given.

Nerve macrophage population

Macrophages were identified by Iba1 labeling. Volumetric segmentations of the spiral ganglion, auditory neuron dendrites within the spiral lamina, and Organ of Corti were applied to Iba1-labeled sample D by converting segmentation volume stack in .nrrd format, extracting ROIs in Fiji with a custom script, and applying the obtained mask to the Iba1 signal stack. Macrophages were then quantified with the 3D Object counter of ImageJ.

Results

The inner ear atlas presented here is based on the combination of high-resolution microCT and iDISCO+ datasets (Fig. 1) acquired from five samples from four adult Wistar rats (Table 1). The atlas features annotations of 34 structures (Table 2), which include 14 novel structures and 20 detailed subdivisions of the inner ear structures currently present in the Waxholm atlas (Kleven et al., 2023). Atlas volumes are registered to the coordinates of the Waxholm rat brain atlas (Fig. 1A) so that volumes derived from both microCT (Fig. 1B-2) and iDISCO+ (Fig. 1B-3) datasets can be substituted in place of the current Waxholm inner ear volumes (Fig. 1B-1) and sectioned on arbitrary planes to reveal annotated structures (Fig. 1B-4).

Figure 1 Organization of the volumetric atlas of the rat inner ear.

(A) Posterior view of a 3D rendering of the Waxholm rat brain atlas v4 (Kleven et al., 2023) displaying its original inner ear (green) on the left side, and our inner ear volumes in the right side (the volumes of all segmented labyrinths -each in a different color- are superimposed in the figure). Inner ear structures were registered to the Waxholm atlas right inner ear (see Methods). (B) The original Waxholm inner ear volumes for cochlea and vestibular apparatus (B1) was registered to labyrinth volumes segmented from microCT scans (B2) and internal soft tissue structures segmented from lightsheet scans (B3), obtaining a volumetric atlas which can be cut on arbitrary planes (B4) yielding reference structures e.g., for histological slices. 8cn: cochlear nerve (d, dendrite; p, peripheral; SpG, spiral ganglion), 8vn: vestibular nerve (i, inferior; s, superior), Art, artery (co, cochlear; st, stapedial; ve, vestibular), Co, cochlea (caq, cochlear aqueduct; OoC, Organ of Corti; slg, spiral ligament; slm, spiral limbus; sm, scala media; st, scala tympani; sv, scala vestibuli; sva, stria vascularis), VeA, vestibular apparatus (aca, anterior canal ampulla; acc, anterior canal crista; asc, anterior semicircular canal; cc, crus commune; ed, endolymphatic duct; er, elliptical recess; lsc, lateral semicircular canal; mut, macula utriculi; p, perylimph; pca, posterior canal ampulla; pcc, posterior canal crista; psc, posterior semicircular canal; rd, reuniens duct; sac, saccule; sr, spherical recess; utr, utricle).

Figure 2 Membranous labyrinth parts.

(A) Vestibular membranous labyrinth volumes from rat A. Semicircular canals (scc) and utricle are connected; the saccule is connected to the remaining vestibular organs indirectly, through the endolymphatic duct. (B) Cochlear fluid spaces from rat B. Scala media is filled with endolymph, scala tympani and vestibuli are filled with perilymph. (C) Around the membranous vestibular labyrinth there is a thin periotic space filled with perilymph (here shown from rat A). (D) Total fluid spaces of rat A inner ear. Saccule and utricle are not visible inside the scala vestibuli. (E) Endolymphatic spaces of rat A inner ear. The labyrinth is divided in inferior (inf, yellow: cochlea, saccule and posterior canal) and superior (sup, lilac: utricle, anterior and lateral canal) divisions.

Figure 3 Reuniting duct.

(A) Volumes from rat B displaying (1) cochlear scala media, reuniting duct, and saccule and (2) same as (1) plus cochlear limbus (lilac) and saccular connective tissue lamina (blue). (B) Single images from rat B stack. Lines in A show the position of the image planes. Scale bar: 400 µm. (C) Volumes from microCT (1) and iDISCO+ (2) segmentation of the periotic cistern region. In microCT scans, the bone located between the spherical recess (sr) and the end of Rosenthal canal (asterisk) displays a tortuous blood vessel, the vestibulocochlear artery (pseudocolored red). In iDISCO scans, a vessel with similar position and tortuosity is found below the ductus reuniens. The bone from microCT scan is added in C2 for context. Whole labyrinth outline is also shown for context. Co, cochlea (rwm, round window membrane; slm, spiral limbus; sm, scala media), sr, spherical recess; TBo, temporal bone; VeA, vestibular apparatus (csa, saccular connective tissue; ed, endolymphatic duct; rd, reuniting duct; sac, saccule).

Figure 4 Geometry and position of the utricle and saccule.

(A) Volumes from rat B. (B) Selected optical sections from lightsheet autofluorescence signal showing different levels of the scala vestibuli periotic cisterna. A1: saccular connective lamina and utricle outline shown superimposed a semitransparent bony labyrinth. Spherical and elliptical recesses mark the position of saccule and utricle, respectively. A2: same as A1, with added scala vestibuli and endolymphatic duct. Horizontal lines show the level of optical sections in B. A3: location of otolithic maculae. Connective tissue structures have been removed, exposing the saccular and utricular maculae. Outside the membranous labyrinth, spaces not filled by connective tissue are filled by perilymph. A4: same as A3, rotated 180 degrees in order to show the membrana limitans attachment to the bony labyrinth. A: anterior; R: right; S: superior. Abbreviations in A and B are from Table 2. Asterisk in B2 indicates utricular connection to bone. Scale bar: 400 µm. 8vn: vestibular nerve (s: superior), Art, sp: artery, stapedial, Co, cochlea (sm, scala media; sv, scala vestibuli), Sp, stapes; VeA, vestibular apparatus (acc, anterior canal crista; csa, saccular connective tissue; ed, endolymphatic duct; lcc, lateral canal crista; ml, membrana limitans; msa, macula sacculi; mut, macula utriculi; p, perylimph; rd, reuniens duct; sac, saccule; utr, utricle).

The atlas terminology is compatible with the inner ear field literature (see below for details), and the terms are hierarchically organized (Table S1). At the highest level of the hierarchy, the inner ear is subdivided into cochlea and vestibular apparatus. Both are in turn subdivided into bony labyrinth, membranous labyrinth, sensorineural structures, and other structures. In addition, volumes for three vascular and two osseous structures have been segmented, because they facilitate recognition and localization of inner ear structures. Detailed 3D structure annotations were created by combining the interpretation of microCT and iDISCO+ datasets with that of partial reconstructions from additional iDISCO samples (see Methods and Table 1), plus microCT and histochemical data from the literature.

Integration of microCT and iDISCO datasets

The bony labyrinth is a set of cavities within the petrosal bone, which opens into the middle ear space through the round window, and into the cerebrospinal fluid space through nerve foramina and the cochlear and vestibular aqueducts (Gulya, 2007). Bony labyrinths from microCT scans (Fig. 1B-2) were easily segmented using active contour segmentation thanks to the high contrast between bone and other substances (air, water and soft tissues). Lugol’s iodine (KI3) counterstaining allowed visualization of the round window membrane and segmentation of the funnel-shaped round window (Goycoolea & Lundman, 1997) in microCT datasets. The round window membrane was also easily segmented in cleared samples, due to its strong autofluorescence (Fig. S3).

Bony labyrinths from lightsheet scans (Fig. 1B-3) were reconstructed from the autofluorescence signal. Fluid spaces of the vestibule and cochlea were visible as dark spaces within the weakly autofluorescent bone, whereas soft tissue structures such as the spiral ligament and round window membrane were strongly autofluorescent. Bony labyrinth outline was therefore reconstructed by joining the outer surfaces of the dark periotic volumes and of the bright spiral ligament and round window membrane. Bony labyrinth segmentations from microCT and lightsheet scans could be registered to each other by rigid transformation (see Methods), indicating that tissue shrinking due to clearing was isotropic. Average Procrustes distance was 0.05 ± 0.01 (n = 5), and the distance between Waxholm and microCT samples (0.05 ± 0.01) was not significantly different from that of lightsheet samples (0.04 ± 0.01).

Since microCT and lightsheet data were obtained from different animals, we could not quantify absolute shrinking; however, tissue shrinking due to the iDISCO+ clearing is known to be moderate: in Renier et al. (2014), shrinking was about 10%. A comparison between cleared and ex vivo CT samples of the gerbil cochlea (Hutson et al., 2021) reported 4.5% shrinking of cleared cochlea vs. ex vivo CT volumes. Volumes for each segmented structure are given in Table 3.

Membranous labyrinth

The membranous labyrinth (Fig. 2) is a set of sacs contained within the bony labyrinth, filled with endolymph and surrounded by the periotic space filled with perilymph (Gulya, 2007). It is divided (Fig. 2E) in an inferior division (cochlea, saccule and posterior canal) and a superior division (anterior and lateral canal and utricle) on the basis of innervation and vascularization (Gulya, 2007). In the cochlea (Fig. 2B), the membranous part is the scala media, sandwiched between the perilymph-filled scala vestibuli and scala tympani. The scala vestibuli ends with an enlarged region which shows the oval window, whereas the scala tympani ends at the round window. Between the cochlea and saccule is the reuniting duct (Smith et al., 2022a); a second thin duct (saccular duct) extends from the saccule towards the utricle expanding into the endolymphatic sinus and then becoming the endolymphatic duct after the utricle (Manni, 1987).

In the vestibular apparatus (Fig. 2A), the three membranous semicircular canals emerge from the utricle, entering the respective bony canals (Fig. 2C); each canal displays at one end an enlarged ampulla carrying the sensory crista. For anterior and lateral canals, the ampulla is close to the utricular macula, whereas the posterior canal ampulla emerges from the nonsensory part of the utricle. Anterior and posterior canals fuse their non-ampullated region forming the crus commune.

All these and other soft tissue structures were segmented from the autofluorescence signal of lightsheet image stacks. Cleared tissue datasets revealed reuniting duct features that, to our knowledge, have not been previously reported and could be important in the communication between cochlear and vestibular endolymph (Smith et al., 2022b). The reuniting duct (Fig. 3) displayed a ribbon shape thinning towards the middle (Fig. 3A), as found in human and guinea pig (Smith, Curthoys & Laitman, 2024). Full reconstruction was possible in two samples (Fig. S2), where its length was 696 ± 6 µm, and minimum width 26 ± 4 µm on the smaller axis and 83 ± 5 µm on the larger axis (see Table 4). The cochlear scala media spirals around the bony modiolus and is anchored on its modiolar side to the spiral limbus, a soft tissue structure. The latter (Figs. 3A, 3B) was found at the cochlear base to expand as a sleeve below the reuniting duct and to attach to the saccule, forming a connective tissue lamina anchoring the saccule to the bone of the spherical recess. In sample B we delineated arteries from their SMA signal, and could recognize cochlear and vestibulocochlear arteries from their tortuous path (Hornstrand, Axelsson & Vertes, 1980; Mei et al., 2018). In the bone below the connective tissue sleeve, the tortuous vestibulocochlear artery (VCA) followed the course of the reuniting duct; a similar tortuous vessel was observed in microCT datasets between the spherical recess, which marks saccular position (Smith, Curthoys & Laitman, 2023) and the end of Rosenthal’s canal, which marks the end of scala media (Fig. 3C). The VCA therefore allows to localize the reuniting duct even in microCT scans, where the duct is not visible.

Table 4 Metrics of reuniting and endolymphatic ducts (mm).

Length and width of thin ducts were measured from high-resolution image stacks of autofluorescence signals; vestibular aqueduct length was also measured from microCT signal.

Sample	RD length	RD minimal width (max;min)	ED saccular duct width	ED sinus width (max;min)	ED length	
A	–	–	0.064;0.024	0.104;0.024	3.16* (2.50)	
B	0.7	0.083;0.023	0.094;0.018	0.138;0.03	–	
C1	–	–	–	–	2.67	
C2	–	–	–	–	2.46	
D	1.08	–	–	–	–	
E	0.69	0.131;0.018	0.083;0.026	0.162;0.036	–	
Average	0.82	0.107;0.02	0.080;0.022	0.134; 0.03	2.55	
St.dev.	0.22	0.011;0.002	0.015;0.004	0.029;0.005	0.11	
Note:

*Cleared samples allowed to follow the endolymphatic duct and sac in their entirety, whereas in microCT samples only the aqueduct and part of the sac are seen. When measuring the segment between the utriculoendolymphatic valve and the operculum (in parenthesis), ED length from cleared sample was similar to that measured from microCT samples (see Fig. 5H). Values in bold and italic represent statistic calculation from samples.

Figure 5 Endolymphatic duct and Bast’s valve.

(A) Medial view of reconstructed volumes of the utricle and saccule displaying their position within the labyrinth and their connection through the endolymphatic duct. (B) Magnification of the volume shown in the yellow rectangle in A, after a 180-degree dotation on its long axis. The utriculoendolymphatic valve is evident as a fold in the duct sinus. (C–F) Orthogonal views of rat B inner ear around the utriculoendolymphatic valve (yellow dot). Scale bar: 400 µm. (E) Inferior view of the endolymphatic duct highlighting the utriculoendolymphatic valve as shown in the orthogonal views. (G) the whole reconstruction of the endolymphatic space, including endolymphatic duct and sac, is superimposed to the rat temporal bone from microCT, in anatomical position. (H) Difference in starting and ending points for measurement of endolymphatic duct and sac in cleared samples (left) and of vestibular aqueduct in microCT samples (right). Co, cochlea (sv, scala vestibuli), VeA, vestibular apparatus (ed, endolymphatic duct; ml, membrana limitans; msa, macula sacculi; mut, macula utriculi; pcc, ppsterior canal crista; sac, saccule; utr, utricle).

Otolithic organs (Fig. 4) are located in the periotic cistern of the cochlear scala vestibuli. The saccule is anchored to the bone of the spherical recess (Figs. 4A, 4B), whereas the utricle is mostly free within the cistern, attached to a connective tissue ledge (Uzun-Coruhlu, Curthoys & Jones, 2007; Figs. 4B, 4C). The otolithic organs are directly in front of the stapes (Figs. 4C, 4D, Video S6) (Smith, Curthoys & Laitman, 2023); consistent with this, they can respond to sound (Young, Fernández & Goldberg, 1977; Curthoys & Dlugaiczyk, 2020). The periotic space around the utricle is filled with connective tissue trabeculae collectively forming the membrana limitans (Smith et al., 2022a), which separates the superior and inferior divisions of the labyrinth and anchors the utricle to the bone wall (Fig. 4D) (Smith et al., 2022a). The periotic space of semicircular canal ampullae displays instead no trabeculae and the ampullary roof is often collapsed in dehydrated samples (including cleared samples) because of volume changes due to dehydration.

Although the membranous utricle and saccule are attached to each other externally, their endolymphatic spaces only communicate indirectly (Fig. 5). The thin saccular duct (Fig. 5A, Table 4) soon joins the enlarged endolymphatic sinus, but the utricle does not show any patent pathway, consistent with the presence of a membranous utriculo-endolymphatic valve (Bast, 1937). The valve (Figs. 5B–5F) had the shape of a vortex-like fold in the endolymphatic duct and may be due to a trabecular link within the periotic connective tissue pulling on one side of the duct. Below the utricular valve, the vestibular aqueduct does not contain perilymph but is filled with loose connective tissue (Hultgård-Ekwall et al., 2003), which could be segmented together with the whole endolymphatic duct and sac, up to the distal sac emerging from bone (Figs. 5G, 5H) Since the first part of the endolymphatic duct and the distal part of the endolymphatic sac are not encased in bone (Dahlmann & von Düring, 1995), duct measures obtained by microCT are shorter that measures obtained in cleared tissue (Fig. 5H).

Sensorineural structures

The VIII nerve (vestibulocochlear, Fig. 6) is formed by a cochlear and a vestibular nerve. The former has its neuronal bodies in the spiral ganglion, located within the osseous Rosenthal canal in the cochlear modiolus (Figs. 6A, 6E). The vestibular, or Scarpa’s, ganglion, is instead located close to the brainstem, and in the rat it is separated in a superior and an inferior division (Fig. 6B). The saccule receives fibers from both divisions (as described in Lindeman (1969)), entering the macula from opposite directions (Figs. 6C–6F) whereas all other organs are only innervated by one division. Nerve fibers and somata in cleared samples were brightly autofluorescent, and therefore nerve fascicles and ganglia could be resolved in both cochlear and vestibular nerves. The spiral ganglion can be traced in both cleared and microCT datasets, since Rosenthal canal is delimited by thin bone walls; however, volumes obtained from microCT are larger, since the whole internal space of Rosenthal canal is segmented (Fig. 6E), whereas in cleared tissue only the autofluorescent neurons are seen (Fig. 6A). Scarpa’s ganglion is not associated to bone and can only be observed on cleared samples. Moreover, in the cleared cochlea it was possible to distinguish central and peripheral myelin (Fraher & Delanty, 1987) by their different fluorescence (Video S1).

Figure 6 Neurosensory structures.

(A) Single optical section from autofluorescence signal in rat B, showing the cochlea, where several soft tissue structures can be seen. In particular, cochlear nerve peripheral zone (8cn, p) displays nerve fascicles, the spiral ganglion (8cn, SpG) displays neuronal somata, and auditory neuron dendrites (8cn, d) display fascicles reaching the Organ of Corti (Co, OoC) through the osseous spiral lamina. Scale bar: 400 µm. (B) Single optical section from autofluorescence signal in rat B, showing the vestibular nerve and its superior and inferior divisions. Saccular sensory epithelium can be segmented from the unlabeled macula thanks to the thickness and autofluorescence of cells. Ganglion cells somata are visible. Scale bar: 400 µm. (C) Reconstruction of the cochlear and vestibular nerve volumes from rat B. Lateral view. Superior and inferior components of the vestibular nerve are separated. (D) Superior view of the semicircular canal cristae and macula utriculi with their nerve supply. (E) microCT image at the level of the lower cochlear turn. Nerves are evident, but the otolithic organs are not delineated. Note that the bony channel encasing the spiral ganglion (Rosenthal’s canal) is visible, but the ganglion itself is not. Scale bar: 400 µm F: Posteromedial view of saccule reconstructed from rat B displaying its double innervation. 8cn: cochlear nerve (d, dendrite; SpG, spiral ganglion; p, peripheral), 8vn: vestibular nerve (i, inferior; s, superior), Co, cochlea (OoC, Organ of Corti; rwm, round window membrane; slg, spiral ligament; slm, spiral limbus; sm, scala media; st, scala tympani; sv, scala vestibuli; sva, stria vascularis), Sp, stapes; VeA, vestibular apparatus (acc, anterior canal crista; lcc, lateral canal crista; msa, macula sacculi; mut, macula utriculi; pcc, posterior canal crista; sac, saccule; utr, utricle).

Although no hair cell-specific labeling was used, sensory epithelia could be traced by autofluorescence, allowing to map the maculae of otolithic organs and the cristae of semicircular canals. In the cochlea, the organ of Corti and spiral limbus were also autofluorescent (Fig. 6A), and basilar membrane length was 10.2 ± 0.3 mm (n = 4), consistent with previous literature data for Wistar rats (Burda, Ballast & Bruns, 1988).

Atlas use example: macrophage population analysis with atlas-defined regions of interest

Atlas volumes can be used to characterize cell populations within inner ear structures. As an example, we extracted the macrophage populations associated with the neurosensory part of the cochlea (Fig. 7). By cropping the cochlear Iba1-labeled stack with the volumes segmented from the autofluorescence signal, macrophage populations associated with the spiral ganglion and spiral lamina dendrites could be isolated (Fig. 7A). With volume segmentation, it was possible to observe the different morphologies of the macrophages located in the spiral limbus, ganglion, dendrites and below the Organ of Corti (Fig. 7B). Macrophages were also seen associated with other cochlear structures (Fig. 7C) but were not characterized. Macrophages within the segmented volumes could be counted (Fig. 7D), yielding a population of 5,839 cells within the neurosensory structures, of which only 28 cells associated to the Organ of Corti.

Figure 7 Volumetric segmentation of macrophage populations in the cochlea.

(A) Maximal intensity projection of the Iba1 signal from the neurosensory volume of the cochlea of rat D (total thickness, 2.5 mm). Scale bar: 1 mm. Cochlear nerve macrophages are shown as yellow on black. The red band around the nerve encompasses the Organ of Corti and part of the spiral ligament. Volumetric model of the neurosensory structures of the cochlea is shown in the bottom left corner: spiral ganglion is orange, dendrites yellow, Organ of Corti magenta. (B) Magnification of the region shown in the white rectangle in A; maximal intensity projection of the Iba1 signal from the segmented neurosensory volume of the apical cochlear turn (total thickness, 0.5 mm). Overlayed colors indicate annotated volumes. Neurosensory structures are shown together, since macrophages populations do not overlap in the maximal intensity projection. The segmented volume of the spiral limbus (purple) from the same cochlear turn is translated vertically for clarity; dashed purple lines indicate its actual position. Note the presence of macrophages with different morphologies in each segmented volume. (C) Single optical slice of the Iba1 signal from sample D, showing the presence of macrophage populations associated to several cochlear spaces. Scale bar: 1 mm. (D) False color rendering of object count from the segmentation in (A). Colors indicate depth. 8cn: cochlear nerve (d, dendrite; SpG, spiral ganglion), Co, cochlea (OoC, Organ of Corti; slg, spiral ligament; slm, spiral limbus).

Discussion

The growing availability of methods for imaging and image analysis has led in the past years to the resurgence of anatomy and generation of digital atlases. For the inner ear, automated segmentation is becoming increasingly precise, but soft tissue presents several ambiguous and intricate regions, where anatomical skill and knowledge, together with hard image boundaries, have to guide region delineation.

The Waxholm Space atlas of the rat brain (Kleven et al., 2023) features a detailed regional analysis of the auditory system (Osen et al., 2019) including five annotated structures in the inner ear. To the latter, we have added 14 novel structure delineations and revised the delineation of 20 structures on the basis of observations made with microCT and lightsheet imaging of cleared temporal bones.

We provide comprehensive descriptions of delineation criteria and demonstrate that histological features of soft tissues obtained from cleared tissue can be integrated with bony labyrinth delineation obtained from microCT scans thanks to morphometrics-based registration. Our bony labyrinth template was acquired in situ within the cranium from a perfusion-fixed specimen, counterstained with iodine (Lugol solution), with a voxel size of (6.23 µm)3. Soft tissues were mostly traced from lightsheet image stacks of iDISCO+ cleared samples, with voxel sizes ranging from (4.08 µm)3 to (2 µm)3. This greatly increases resolution vs. that of sMRI-based volumes, where resolution was 39 µm. Lightsheet imaging allowed the reconstruction of thin, morphologically complex soft tissue structures such as the reuniting and endolymphatic ducts and utriculoendolymphatic valve.

We have, to our knowledge, provided the first description of the rat reuniting duct association with the cochlear spiral limbus and vestibulocochlear artery. This description may help the study of communications between cochlea and sacculus, since the artery path through bone is visible with microCT and therefore can provide a reference point for the reuniting duct in microCT imaging. Moreover, the reuniting duct association with the spiral limbus and saccular connective tissue may explain differences of immune responses between the cochlea and vestibular system, since in the cochlea lymphocytes infiltrate the organ from the modiolus (where the limbus resides) and neutrophils from the spiral ligament, whereas the vestibular organs do not display separate pathways (Keithley, 2022). Further study of the changes of the reuniting duct and its associated immune responses in conditions of inflammation or blockade may shed light on the mechanisms involved in hydrops spreading from the cochlea to the vestibular system, where the correlations between clinical findings and animal experimentation are still unclear (Álvarez De Linera-Alperi et al., 2024). Given the thin lumen of the duct, local inflammation or otolith clusters could be able to transiently close it, thus sealing the cochlea from endolymph flow, potentially triggering cochlear hydrops without a vestibular involvement.

Lightsheet fluorescence-based imaging displays some differences from x-ray-based imaging. In particular, although the mesoSPIM employs ASLM to achieve uniform z-axis resolution across the field of view (Voigt et al., 2019), lightsheet image stacks are not isotropic, since they are affected by z-axis distortion. In order to make our datasets isotropic, we performed a post-imaging deconvolution of lightsheet datasets (Becker et al., 2019) and increased xy-pixel size to match z-axis. As a clearing technique, we employed iDISCO+, a solvent-based clearing technique optimized for immunolabeling, which induces a modest sample shrinking, of the order of 10% (Renier et al., 2014), and allows the joint observation of inner ear and brain (Perin et al., 2019). Previous studies in the cochlea suggested that, after decalcification, bone tissue shrinks 8.4% similarly to soft tissues (Buytaert et al., 2011). Moreover, Lugol’s iodine (KI3) has been found to induce tissue shrinking around 15–20% (Bogoevski et al., 2019; Costello et al., 2024). Although we did not obtain microCT and lightsheet images from the same samples, the morphology obtained with the two techniques were well congruent after rigid transformation, therefore suggesting low variance in the adult rat population, despite the presence of important age differences between animals.

Membranous semicircular canals appear to be the weakest part of the atlas, since decalcified cleared bone becomes deformable, and therefore semicircular canals arms do not perfectly follow their CT counterparts, displaying the largest Procrustes distances among landmarks (see Fig. S1). Moreover, collapse of ampullary roof is a common occurrence in PFA fixed, dehydrated samples, most likely due to the ampulla shape (a sphere is easily deformed) and scarcity of connective tissue trabeculae anchoring it to the bony labyrinth. Further experiments with a stronger fixative (Anniko & Lundquist, 1977) will address the issue.

The autofluorescence of cleared tissue allows tracing of soft tissues even without immunolabeling (Weiss et al., 2021). In fact, the animals used in the present study were labeled for blood vessels or immune cells, but only autofluorescence was used for segmentation. Although we did not label sensory cells, the sensory epithelia were distinguishable thanks to their higher fluorescence and thickness vs. the surrounding tissue. The outline of utricular and saccular maculae segmented using these cues was similar to those in freshly dissected organs (Eatock et al., 1998; Wooltorton et al., 2007; Songer & Eatock, 2013). On the other hand, since lightsheet imaging is not confocal and suffers from halo artefacts (Weiss et al., 2021), it was not possible to reliably measure the wall thickness of membranous structures even in our highest resolution samples.

One of the strengths of tissue clearing for the inner ear is the immunolabeling of cell populations in situ, by employing regular antibodies (Risoud et al., 2017; Perin et al., 2019) but also protein constructs smaller than antibodies, such as nanobodies (as used with vDISCO, Cai et al., 2023) or conjugated antibody fragments (as in Baart et al. (2023)). An example of volumetric analysis is shown in the present article, where cochlear macrophages were labeled with Iba1, and volumetric segmentation was applied to isolate macrophage populations associated with the cochlear nerve and sensory epithelium. Results of this analysis show that macrophage density in the spiral ganglion and dendrite (almost 4,000 cells/mm3) is within the range of microglial cell density in the brain (Keller, Erö & Markram, 2018), whereas only 28 macrophages were found in association to the Organ of Corti throughout the whole cochlear spiral. Moreover, spiral limbus macrophages displayed morphological differences (thicker and less elongated processes) when compared to nerve-associated macrophages. Further studies will help characterize these different populations, which have been found to play different and very important roles (Warchol, 2019).

Interpretations based on the present atlas should consider the methodological limitations outlined above for tissue clearing, and also note that the atlas is based on four animals, rather than a population average. Although physiological properties are known to change with age (Borg, 1982; Keithley, Ryan & Feldman, 1992; Buckiova, Popelar & Syka, 2007; Cai et al., 2018), at our level of detail no significant morphological differences were seen between young (3 months) and older (15 months) animals. Spiral ganglion volume, which was found to be reduced in old mice (Santi & Johnson, 2022), was not age-dependent in our samples, although sample size is too low for statistical considerations. Moreover, our samples were obtained from Wistar rats, and although morphology of the auditory system appears to be generally consistent in Sprague-Dawley, Wistar and Long Evans rats (Osen et al., 2019), minor differences may be present, which were not appreciated at our level of detail (Buckiova, Popelar & Syka, 2007).

Despite these limits, it has to be remembered that the volumetric reconstruction and curation of a novel or complex structure still requires a large amount of manual work. In the inner ear, although automated segmentation is becoming increasingly precise, soft tissue presents several ambiguous and intricate regions, where anatomical skill and knowledge, together with hard image boundaries, have to guide region delineation. A fundamental contribution of the present rat inner ear atlas is that the atlas delineations are publicly shared and can be easily registered to the Waxholm rat brain atlas. Although the limited number and heterogeneity of samples used reduces at the moment the atlas precision, the open access and multimodal nature of the atlas will easily enable the incorporation of novel data.

Supplemental Information

Supplemental Information 1 Labels for volume segmentations.

Abbreviations are in Table 2.

Supplemental Information 2 Landmark coordinates for registration.

Coordinates for the landmarks used in registrations generated by SlicerMorph as .fcsv files.

Supplemental Information 3 Transform matrices for affine transformations.

Transform matrices for affine transformations generated by SlicerIGT as .h5 files

Supplemental Information 4 Ontology of inner ear structures.

The inner ear is divided in Cochlea and Vestibular Apparatus, following (Osen et al., 2021). Each subdivision is divided into bony labyrinth, membranous labyrinth, sensorineural and other structures. Arteries and bones are classified in a separate "other" category.

Supplemental Information 5 Bony labyrinth registration.

A: Ellipsoids showing Procrustes distances (magnified 5x) for each landmark placed on the labyrinths. B: Superposition of bony labyrinth volumes from microCT and lightsheet samples after rigid transformation (scaling, rotation, translation) for maximal congruence with ALPACA.

Supplemental Information 6 Reuniting duct volumes from rat E2 (left) and B1 (right).

The saccule in rat E2 was only partially visible, and its reconstruction is incomplete. Rd: reuniting duct; ed: endolymphatic duct

Supplemental Information 7 Round window region from cleared (left) and microCT (right) sample.

A: single optical section of Rat B showing autofluorescence signal. Scale bar: 1 mm. B: single section of microCT stack displaying bone and several soft tissue elements, such as the round window. Scale bar: 1 mm. C, D: same as A, B but with annotated segmentations added. 8cn: cochlear nerve, 8vn: vestibular nerve (i: inferior, s: superior), BoL: bone labyrinth, Co: cochlea (rwm: round window membrane, slg: spiral ligament, slm: spiral limbus, sm: scala media, st: scala tympani, sv: scala vestibuli), Sp: stapes, TBo: temporal bone, VeA: vestibular apparatus (ed:endolymphatic duct, mut: macula utriculi, p:perylimph, rd: reuniting duct, sac: saccule, utr: utricle).

Supplemental Information 8 Rat A volumes progressively shown with colors as in Table 2.

Supplemental Information 9 Rat A membranous labyrinth volumes.

Supplemental Information 10 Rat B volumes, nerve to scala media.

Progressively shown with colors as in Table 2, part 1.

Supplemental Information 11 Rat B volumes, endolymph and epithelia.

Progressively shown with colors as in Table 2, part 2.

Supplemental Information 12 Rat B volumes, perilymph and other.

Supplemental Information 13 Vestibular sensory epithelia and nerves from rat B.

Supplemental Information 14 Temporal bone with membranous labyrinth.

Semitransparent volume of rat temporal bone from sample C, superimposed with membranous labyrinth from sample A (brown).

Supplemental Information 15 Endolymphatic duct with Bast’s valve.

Volumetric reconstruction of the initial part of the endolymphatic duct displaying Bast’s valve. Same view as in Figure 5B.

Additional Information and Declarations

Competing Interests

The authors declare that they have no competing interests.

Author Contributions

Daniele Cossellu performed the experiments, analyzed the data, prepared figures and/or tables, authored or reviewed drafts of the article, and approved the final draft.

Elisa Vivado performed the experiments, analyzed the data, prepared figures and/or tables, authored or reviewed drafts of the article, and approved the final draft.

Laura Batti performed the experiments, analyzed the data, prepared figures and/or tables, and approved the final draft.

Ivana Gantar performed the experiments, analyzed the data, prepared figures and/or tables, and approved the final draft.

Roberto Pizzala conceived and designed the experiments, analyzed the data, authored or reviewed drafts of the article, and approved the final draft.

Paola Perin conceived and designed the experiments, performed the experiments, analyzed the data, prepared figures and/or tables, authored or reviewed drafts of the article, and approved the final draft.

Animal Ethics

The following information was supplied relating to ethical approvals (i.e., approving body and any reference numbers):

The study was approved by the Italian Ministry of Health (protocol number 155/2017-PR). The study was approved and supervised also by the University of Pavia Animal Welfare Office (OPBA) together with the Board of the local animal house core facility with protocol #PT117. This study was carried out in accordance with the recommendations of Act 26/2014, Italian Ministry of Health.

Data Availability

The following information was supplied regarding data availability:

The inner ear segmentations and accompanying label descriptions for atlas integration are available as standard NIfTI and label files (both original _ORG and transformed _WXS), which can be visualized with ITK-SNAP (Yushkevich, Gao & Gerig, 2016). Segmented and annotated volumes for 3D displaying, editing and printing are available as .ply files. Original scans are shared as _grayscale.nrrd stacks. All are available at Zenodo: Cossellu, D., Vivado, E., Pizzala, R., & Perin, P. (2025). Volumetric atlas of the rat inner ear from microCT and iDISCO+ cleared temporal bone. Zenodo. https://doi.org/10.5281/zenodo.14866967.

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
