# Peer review of "Volumetric atlas of the rat inner ear from microCT and iDISCO+ cleared temporal bones"

_PeerJ, doi:10.7717/peerj.19512_

## Round 0.1 · original submission · Major Revisions

Dear authors,
It seems the manuscripüt needs more work. Especially reviewer 2 phrases it and also reviewer 1. Please make sure that you follow their advice when preparing a revised version. Also, the points reviewer 3 raises need attention. Overall that should make the manuscript shorter, more compact and more "atlas-like" and would increase the usability as I understand.

**Language Note:** The review process has identified that the English language must be improved. PeerJ can provide language editing services - please contact us at [email protected] for pricing (be sure to provide your manuscript number and title). Alternatively, you should make your own arrangements to improve the language quality and provide details in your response letter. – PeerJ Staff

Reviewer 1 ·

Basic reporting

Major comments:
This is a rather well-written manuscript on iDISCO-cleared temporal bone. The authors have utilized a tissue-clearing method and light-sheet microscopy to analyze decalcified temporal bones of three adult Wistar rats, 1 male and 2 females, aged P494, P92, and P111, respectively. They investigated the dimensions of both the soft and hard tissues of the inner ear labyrinth, and some of the material is posted online as an anatomical resource. In the Introduction, the authors have clearly stated the need for a detailed morphometric analysis of inner ear using modern imaging methods. They would like to relate their results to the Waxholm Space atlas, a well-known neuroscience resource, which currently contains a detailed analysis of the rat auditory brainstem, but no inner ear. The reviewer wonders, however, about the potential utility of the current results, however, which is somewhat dubious, since this purports to be an “atlas”, yet no coordinates that would relate these results to the existing atlases are given. It would have been awesome to also have either some movies, or an interactive 3D model, as part of the supplementary material, which was kind of dark and very pixelated. Some parts of the manuscript could be clearer and there are several grammatical errors (see Minor comments). Overall, however, the results are interesting from the standpoint of relating the morphology of both the soft and hard tissue parts of the inner ear. Most other morphometric studies merely scan fossil or museum specimens of temporal bones with microCT, which doesn’t allow for visualization of soft tissue elements.

General grammatical comments:
The authors use “connective” at many places within the manuscript, when I think they mean “connective tissue”. They were too numerous to list, but a search should reveal all of them.

Certain phrases should be set apart by commas, for example, phrases with “for example” or “e.g.” in them.

Some sentences are just too long. For example, the sentence/paragraph beginning on L. 56 could be broken into at least two, such as “…(OPFPS). More recently, tissue clearing has become…”. Also, the sentence on L. 290 beginning with “Our data…” could be divided into at least three or four sentences by getting rid of the colons.

References: Pick one style (Title Case, or not), and choose whether to have https links or doi: links. Some have hyperlinks and some don’t. Please make the style consistent throughout.

There are a few examples of adverbs being used as adjectives, for example, on L. 80 and L. 213, “consistently” should just be “consistent”.

Minor comments (Line numbers, as given by the authors):
LL. 16-17. In the Abstract, the phrase “…for this animal, at difference from other…”, I think it should read either “…for this animal, a difference from other…” or “…for this animal, in contrast to other…”.

L. 61-62. Can the authors give some examples of the commercial and open-source image analysis programs they are advocating?

LL. 80-81. “Lee et al. 2010”.

L. 100. Period missing between “Health” and “The”. Same on L. 110, a period is missing.

L. 112. Suggest “…with a change every hour…”.

L. 115. Put a comma after Abcam.

L. 117. What is Lugol immersion?

LL. 127-128. What is an “AI filter”, and does “px” stand for pixels?

L. 141. If the images were spatially registered within the Waxholm Space template, where are the coordinates?

L. 145. What is a “Procrustes measure”?

L. 162. Suggest adding “the” after “…which enhances segmentation of the…”.

L. 169. By “CNS liquoral space”, are the authors referring to the “cerebellopontine angle cistern” or the “cerebrospinal fluid” or the “subarachnoid space”?

L. 170. What is “snake segmentation”?

LL. 173-174. “…although it did not allow us to resolve…”. Also, “(see, e.g., Fig. 8E).”

L. 175. Need a space between Fig. and 1B.

L. 180. Change to “…tissue shrinking due to the iDISCO+ clearing method is known…”.

L. 184. Change to “…(Figs. 2,3). The membrane labyrinth…”.

L. 193. Suggest changing to “…not empty, but rather trabecular to varying degrees. These trabeculae form…”. In fact, “trabecules” is used several times throughout the MS and is not a standard anatomical term. It’s better to use “trabeculae”. Please change all instances to “trabeculae”.

L. 204. Suggest adding this “Young et al. 1977” citation for vestibular organs and nerve responding to acoustic stimuli. Young ED, Fernández C, Goldberg JM. Responses of squirrel monkey vestibular neurons to audio-frequency sound and head vibration. Acta Otolaryngol. 1977 Nov-Dec;84(5-6):352-60. doi: 10.3109/00016487709123977. PMID: 303426.

L. 206. Harold Schuknecht died in 1996. I think the proper citation for this book would be Merchant et al. 2010, which changes the alphabetical order and numbers of the References.

LL. 209-210. “Utricle and saccule only communicate indirectly…”. L. 213, “consistent”

L. 216. “…filled with connective tissue…”, not “connective”. Search for all instances, as mentioned above in general comments.

L. 225. Change to “…separated into a superior and inferior …”. Also, check LL.

L. 226. Lindeman (1969) and others have also noted the dual innervation of the saccular macula.

L. 226. “Utricular innervation derived from the bony attachment and coursed within the connective tissue ledge.” How can innervation be derived from bone? Can you clarify?

L. 264. “…in future studies,…”

L. 267. “…reuniens and endolymphatic ducts...”.

LL. 279-280. “…course through the thin bone around the neck of the paraflocculus, and…”

L. 281. “…or individual variability.” This phrase is a bit lost and was not comprehensible until I read it several times. Suggest reversing the sentence and putting the parenthetical information into a separate sentence.

L. 290. “Our data allowed us to reconstruct…”.

L. 315. “…smaller protein constructs, such as nanobodies…”?

L. 326. “… limitations of the present study are the variable…”, not “regard”.

LL. 333-334. “… thank our families, who endured us working during the holidays.” Personally, I think you should just thank your families and take out the comment about being forced.


References
Please pick one style, all Title Case or not?

Ref. 11. Take out “Volume”.
Ref. 28. Update?
Ref. 41. No volume number, take out “Issue”.
Ref. 44a? No number for the reference between 44 and 45.
Ref. 58. Again, the proper citation is Merchant et al., so reorder this one alphabetically. List chapter and page numbers for the chapter referenced.
Ref. 66. Should be “Nature Reviews Neuroscience”
Ref. 70. Needs a URL or some other way to access.
Refs. 74 and 75. Make style consistent. No first names, just initials.


Figures
Fig. 1. Which rat is this? Also, Col IV should be defined. Probably also CT.

Figs. 2 and 3. The colors are listed in the legend, as well as their codes being listed in Table 1.

Fig. 4. “rototranslation”? Also, put period after “deformation”.

Fig. 5. “Blood vessels are evident…”? Not really, please indicate some of them with arrows. No scale bar is evident. What do the nearly-invisible green arrows point to? Please define.

Fig. 6. This legend is very poorly written. The abbreviations for structures are not defined in the legend, as they should be both here and in the Table. There should be arrows and the word “arrows” indicating things like the “trabeculae (thin arrows)” and the “connective tissue ledge (thick arrows)”. In panel B, there is no saccular macula labeled, yet the legend says “The saccular macula is attached…”. For panel C, the legend says “The bulla in Fig. 1 is shown…”. What does this mean? Fig. 1 or panel C1? If the latter, then those four panels should be labeled. Several abbreviations are not defined.

Fig. 7, legend. Suggest “The flat lumen of the ductus reuniens is surrounded…”. Several abbreviations are not defined.

Fig. 7. Panels B1-3 need to be labeled as such, as in panels E1-3. For D2, I suggest making the label “black-on-white” to make it stand out. The black square in D2 is not visible at the page size. Suggest the same black-on-white or a white square, since the black-on-gray is not working.

Personally, I would have liked to see more images like Fig. 7F, with the soft tissue structures visible within the transparent bone. I would also like to be able to manipulate them, which is somewhat possible with the current supplementary material, although the resolution of the supplementary material is very pixelated.

Fig. 8, legend. A. Where are the cochlear nerve fascicles? Can you indicate them with arrows? B. In a similar fashion, in panel B, the vestibular nerve and ganglion are not indicated. “epithelium” is misspelled. The scale bar is not visible. The abbreviations need to be defined in the legend and some arrows should be deployed. Finally, there’s a missing period at the end of the legend.
Fig. 8. I would lose the blue background in panels C, D and F. Particularly in Panels C and D, it’s very difficult to see the light pink membranous structures against the blue background.

Table 1 legend. “color” should be “colors”. “legend” needs a period after it. I would also suggest using “Color codes” instead of “Color names”, since the hex numbers are a type of code.

Table 1. Don’t forget to add “tissue” to most instances of “connective”, as in “saccular connective tissue”.

Experimental design

This is original primary research, falling within the Aims and Scope of the journal, and the methods are described with sufficient detail and information to replicate. The use of some kind of coordinate system, as in most atlases, would make it more generally useful.

Validity of the findings

The data are robust. Conclusions are well-stated.

Reviewer 2 ·

Basic reporting

This paper presents a 3D atlas of the rat inner ear, comprising segmentation of 20 inner ear structures from both light-sheet and micro-CT datasets. While a meticulously segmented and annotated 3D dataset of the rat inner ear holds potential value for the community, the current paper primarily offers raw data, necessitating further technical analysis and investigation to be beneficial. Notably, the absence of quantitative analysis on the datasets and the paper's convoluted structure reduces the enthusiasm of its overall utility. The discussion lacks coherence, largely due to the absence of quantitative results.

Experimental design

Key Issues:
1. The atlas includes data from only three animals, imaged using two different setups, introducing variability that requires discussion, particularly regarding the potential influence of animal age on the obtained structures.
2. The shared datasets should include not only raw images but also segmented organs with clear annotation, as currently, only raw TIFF files are provided.
3. Quantitative data such as organ volume per age and membrane thickness, essential for various applications including inner ear fluid simulations, are missing and should be incorporated.
4. Figure 4 lacks quantitative comparison, making its value unclear. Authors should provide estimates for tissue shrinkage due to clearing.
5. Correcting the claimed resolution that it will be in line with the Nyquist rate. If the pixel size is of 4 µm, the resolution should be at least 8 µm.
6. In line 34 of the abstract, it’s said that “Microwave-aided decalcification yielded sharper images”, which was not discussed at all in the main text.
7. Figures lack sufficient detail and clarity; abbreviations should be provided in the legends, and the details in figures 1 and 2 are too small to be informative.
8. Introduction: The authors need to provide strong motivation how the Atlas or more precisely how their datasets will be used. The introduction abruptly ends, and the impact of their work is not elaborated. Additionally, the unique role of the rat model in auditory research was not sufficiently illustrated in the introduction. How would the knowledge gap be filled by this research given the existing Vaxholm Space atlas of the Sprague Dawley rat brain (from line 81 to line 90)?
9. The methods section needs more detail on the segmentation process for inner ear structures, including evaluation of segmentation accuracy and the software used for 3D reconstruction.
10. The discussion should be better organized, focusing on coherent topics and comparisons to previous works, by generating quantitative data this issue will be resolved.
11. In line 141, it’s said that the images were registered to the Waxholm Space template. What was this step for?


Minor Points:
1. In Figure 4, what’s the green color?
2. In Figure 5, is A showing rat A or a different sample?
3. In Figure 6, do C and D show the cleared inner ear same as A and B? In general, sometimes it is unclear which sample/dataset (light-sheet vs micro-CT) is shown in the figures.
4. Scale bars are missing in Figure 5, Figure 6, Figure 7, Figure 8.
5. (Fitzpatrick et al. 2021) in line 277 is not found in the reference list.
6. Use of words could be improved, e.g., line 251 “goodness of AI training”, line 256 “Together with clinical practice and research”.
7. A few sentences are too long, e.g, “Improvement of X-ray…” from line 44 to line 51, “Tissue clearing of …” from line 56 to line 62.
8. Sentences to rephrase: line 249 “Manual or supervised, semiautomated segmentations thus obtained yield datasets which may subsequently used for AI training”; line 254, “Even so, high-resolution details imaged by software …”; line 273 “As a clearing technique, we employed iDISCO+ …”.
9. In line 88 it should be “voxel size” instead of “pixel size”. Line 257: “ongoing” instead of “ungoing”.

Validity of the findings

The absence of quantitative data presents a significant concern. Without such data, it becomes challenging to draw meaningful conclusions. The lack of quantitative analysis undermines the authors' ability to substantiate their findings effectively.

Reviewer 3 ·

Basic reporting

The article provides an interesting and novel way to visualize minute membranous structure of the inner ear. The authors provide sufficient background and context for the investigation. Article structure is clear and all figures and tables are very useful. I commend the authors in uploading the 3D meshes to Morphosource. This is an excellent addition to the manuscript.

However, I have two comments on the manuscript in its current state. The writing is not entirely clear in some areas and English grammar needs some improvement. These specific areas are described below in "Additional comments". Please also check typos as several have been noted throughout. Also, most figures are lacking scale bars and abbreviations should be defined in the corresponding figure legends.

Experimental design

To the best of my knowledge, this article is original research and fits the scope of PeerJ. Visualization of inner ear structure continues to be a challenge. Therefore, this study contributes greatly to filling a knowledge gap in the literature. The aim of the study is also clearly laid out. The methods are described in detail and are replicable. The experiments have been carried out to conform with the ethical standards of the Italian Ministry of Health.

Validity of the findings

Conclusions are clear and support results. However, the authors could also state that raw 3D landmark coordinates are available upon reasonable request of the corresponding author.

Additional comments

In addition to the comments above, I have several specific comments for revision:

• “Membrane labyrinth” is used throughout but should be “Membranous labyrinth”
• “Connective” is used throughout where I believe “connective tissue” is meant. Please review and revise where needed.
• Line 17: “at difference from other mammals” should be “which differs from other mammals”
• Line 21: Can you please define iDISCO+ as it is the first time used here.
• Line 27: “Cleared bony labyrinth” should be “The cleared bony labyrinth”
• Line 120: Please define “deconvolved” as the audience may not be familiar with the term.
• Line 145: Please give a brief explanation of Procrustes measure. And also please clarify what exact measure is used here. For example, are you examining Procrustes distances among landmark coordinates?
• Line 168-169: Please define what you mean by “CNS liquoral space” I assume this is referring to the subarachnoid space filled with cerebrospinal fluid? Please clarify.
• Line 184: “Figg” should be “Fig.”
• Line 194: Should “periotic connective” be “periotic connective tissue”?
• Line 202: “In facts” should be “In fact”.
• Line 209: I believe the “free end” of the canal is more commonly referred to as the “slender end”.
• Line 216: “but is filled with connective…” please specify what connective material you are referring to here.
• Line 217: This line seems out of place “Collagen IV labeling allows the segmentation of the whole endolymphatic duct and sac, up to the distal sac emerging from bone (Fig. 7F).” Perhaps consider moving it up further into the intro of the paragraph or into Methods.
• Lines 290-299: The section beginning with “Our data allowed to reconstruct…” is one long run-on sentence. Please revise into separate concise sentences.
• Is there a reason why only rat A and rat B are shown in Figures 2 and 3?
• If possible, I would enlarge the images in Figure 2 and 3. The individual structures are very hard to see.
• Figure 4, A and B are noted in the legend but not included in the figure itself. Please add “A” and “B” to the figure.
• Figure 6: I am not familiar with the term “vestibular cisterna”. Do you mean the vestibular caecum? Please clarify/define what you mean and add the label to the figure.
• Figure 8: In the legend, “epitelium” should be “epithelium”.

Supplementary information

• I only see one table and one image, but the document is labeled TableS2. I think these should be labeled Table S1 and Figure S1. If changed, please include the table and figure legend on the word document.

---

## Round 0.2 · Minor Revisions

Please consider the last minor comments of the reviewers.

Reviewer 2 ·

Basic reporting

The authors have performed an excellent revision, well done.

For increased clarity, I recommend addressing the following points:

1. Supplement the figure legends with the inner ear structure abbreviations found in Supplementary Table 1. Many current annotations, particularly 'CO' and 'OCO' for the Organ of Corti, are not immediately clear to the reader.

2. Correct the units associated with numerical values. Specifically, voxel volumes should be reported in cubic micrometers (μm³), not micrometers (μm).

Experimental design

The authors have performed an excellent revision.

Validity of the findings

The authors have performed an excellent revision, well done.

Reviewer 3 ·

Basic reporting

The authors have done substantial revisions to improve the manuscript and have sufficiently addressed my previous concerns. I only have minor suggestions for improvement in the main text:

Line 20: Please explain what exactly is meant by “coarse inner ear”?
Line 22: I would recommned replacing “in” with “with”
Lines 23-24: I would rephraase to say: We performed iodine-enhanced microCT and iDISCO+-based clearing and fluorescence lightsheet microscopy imaging on a sample of rat temporal bones.
Line 30: What about the image resolution? Was it high-resolution?
Line 541: Please define “PSF”
Line 1604: …I would rephrase the end of the sentence to something like: “…will easily enable the incorporation of novel data.”

Experimental design

Study design is clear and the authors have now provided enough information to replicate easily.

Validity of the findings

Results and conclusions are well stated and supported by data.

---

## Round 0.3 · accepted · Accept

Dear Dr. Pizzala and Dr. Perin,

The manuscript is now ready to be accepted.

Yours
Clara Stefen